# Direct Z-Scheme Heterojunction Catalysts Constructed by Graphitic-C_3_N_4_ and Photosensitive Metal-Organic Cages for Efficient Photocatalytic Hydrogen Evolution

**DOI:** 10.3390/nano12050890

**Published:** 2022-03-07

**Authors:** Chuying Lv, Su Qin, Yang Lei, Xinao Li, Jianfeng Huang, Junmin Liu

**Affiliations:** 1The Key Laboratory of Low-Carbon Chemistry & Energy Conservation of Guangdong Province, School of Materials Science and Engineering, Sun Yat-sen University, Guangzhou 510006, China; lvchy3@mail2.sysu.edu.cn (C.L.); leiy56@mail2.sysu.edu.cn (Y.L.); lixao@mail2.sysu.edu.cn (X.L.); 2School of Chemical Engineering and New Energy Materials, Zhuhai College of Science and Technology, Zhuhai 519041, China; qins3@mail2.sysu.edu.cn

**Keywords:** graphitic carbon nitride, metal-organic cage, direct Z-scheme photocatalyst, H_2_ evolution, water splitting

## Abstract

The demand for improving the activity, durability, and recyclability of metal-organic cages (MOCs) that work as photocatalytic molecular devices in a homogeneous system has promoted research to combine them with other solid materials. An M_2_L_4_ type photosensitive metal-organic cage MOC-Q2 with light-harvesting ligands and catalytic Pd^2+^ centers has been synthesized and further heterogenized with graphitic carbon nitride to prepare a robust direct Z-scheme heterojunction photocatalyst for visible-light-driven hydrogen generation. The optimized g-C_3_N_4_/MOC-Q2 (0.7 wt%) sample exhibits a high H_2_ evolution activity of 6423 μmol g^−1^ h^−1^ in 5 h, and a total turnover number of 39,695 after 10 h, significantly superior to the bare MOC-Q2 used in the homogeneous solution and the comparison sample Pd/g-C_3_N_4_/L-4. The enhanced performances of g-C_3_N_4_/MOC-Q2 can be ascribed to its direct Z-scheme heterostructure, which effectively improves the charge separation and transfer efficiency. This work presents a rational approach of designing a binary photocatalytic system through combing micromolecular MOCs with heterogeneous semiconductors for water splitting.

## 1. Introduction

It is meaningful but challenging to convert solar energy into chemical energies and to produce useful fuels such as H_2_ and CH_4_ [1,2,3]. Inspired by enzymatic photosynthesis in nature, scientists have attempted to develop photochemical molecular devices (PMDs) that can conduct catalysis reactions under appropriate conditions, and have adjustable reactivity and properties as well as highly researchable structure—activity relationships [4,5,6]. Metal-organic cages (MOCs) are a kind of discrete cage compound assembled by organic ligands and metal ions, which are named metal-organic polyhedrons (MOPs), macrocycles, containers, cages, and so on [7,8]. In particular, photosensitive MOCs that involve light-harvesting ligands and redox-active metal cation centers can transfer exciting electrons from ligands to catalytic centers fast, and thus work as PMDs, attracting extensive interest from researchers in the field of visible-light driven H_2_ production [9,10,11,12].

However, most photocatalytic MOC systems suffer from easy deactivation, are difficult to recycle, and have poor stability in water [13]. To overcome these drawbacks, efforts have been paid to developing composite catalysts formed by MOCs and some other solid materials for improving the photocatalytic performance and durability of MOCs. For example, Su and co-workers successfully heterogenized an as-synthesized MOC (MOC-16) by embedding it into a metal-organic framework (MOF) named ZIF-8, obtaining a heterogeneous PMD, MOC-16@CZIF [14]. MOC-16@CZIF inherited the efficiently directional electron transfer property of MOC-16, and the MOF matrix provided a platform for proton transfer. Therefore, the H_2_ production activity and durability of the MOC-16@CZIF was much higher than homogeneous MOC-16.

Since Wang and co-workers first used graphite phase carbon nitride (g-C_3_N_4_) for photocatalytic H_2_ production [15], g-C_3_N_4_ has become a widely investigated photocatalyst [16,17,18,19,20,21,22,23,24], because it is a non-toxic, low-cost, easily-prepared, and stable metal-free polymer with an appropriate energy band structure to absorb visible light and is used for many redox reactions. Nevertheless, the bare g-C_3_N_4_ has some disadvantages of a poor visible light response, low charge transfer rate, and low electron-hole separation efficiency, limiting its application in photocatalysis. Therefore, various strategies, such as morphology regulation, metal/non-metal element doping, dye sensitization, co-catalyst loading, and the formation of type II or Z-scheme heterojunction [25], have been investigated. Among these heterojunctions, type II heterojunction systems are widely used, but the photogenerated electrons in the type II heterojunction tend to accumulate on the more positive conduction band (CB), while the holes are prone to gather on the more negative valence band (VB), thus the redox activity is natively weakened thermodynamically. In contrast, the Z-scheme system [26], which simulates the electron transfer process of between photosystem I and photosystem II in nature, reserves both the higher excited energy level and the lower oxidized energy level, and thus shows a stronger redox ability than the traditional type II heterojunction [27,28].

Our group designed a non-photosensitive MOC-PC6 and a photosensitive MOC-Q1, and combined them with semiconductor matrixes including TiO_2_ [29] and g-C_3_N_4_ [30,31] for enhancing their performances. In 2021, we synthesized a M_2_L_4_ type photosensitive metal-organic cage MOC-Q2 with light-harvesting ligands and catalytic Pd^2+^ centers (Figure 1). MOC-Q2 has been successfully heterogenized through combining it with TiO_2_ semiconductor through a simple sol−gel method. The as-prepared MOC-Q2-TiO_2_ photocatalyst showed a dramatically higher activity than the bare MOC-Q2 in a homogeneous solution [32]. To further enhance the performances of the MOC-Q2-based PMD, we herein designed a new binary Z-Scheme heterojunction system constructed with g-C_3_N_4_ and MOC-Q2. The optimal g-C_3_N_4_/MOC-Q2 (0.7 wt%) catalyst achieved a remarkable H_2_ evolution efficiency, giving a production rate of 6423 µmol g^−1^ h^−1^ after 5 h and a turnover number (TON) value of 39,694 based on MOC-Q2 after 10 h, representing the highest of all of the reported MOC-based PMD under visible light irradiation.

## 2. Materials and Methods

### 2.1. Materials

The solvent acetonitrile utilized in the cyclic voltammetry test was an HPLC grade reagent, and dimethyl sulfoxide was dried with A4 molecular sieves. All the other chemicals and reagents were purchased from commercial channels. Unless otherwise specified, further purification was unnecessary before use.

### 2.2. Characterizations

Powder X-ray diffraction (PXRD) curves were tested under Cu-Kα1 radiation (λ = 1.54056 Å) in a Rigaku Smart Lab diffractometer with Bragg−Brentano geometry (Rigaku corporation, Tokyo, Japan). Transmission electron microscopy (TEM) and element mapping were recorded by a JEM-ARM200P spherical aberration corrected transmission electron microscope (JEOL Ltd., Tokyo, Japan). Scanning electron microscope (SEM) images came from Hitachi ultra-high-resolution FE-SEM SU8010 (Hitachi High-Technologies Corporation, Tokyo, Japan). The UV−VIS absorption spectra were obtained on a Shimadzu UV-3600 spectrometer (Shimadzu Corporation, Kyoto, Japan). The infrared spectra were measured by a Frontier Fourier transform infrared (FT-IR) spectrometer with an average of 64 scans in a spectra range of 4000–450 cm^−1^ (PerkinElmer, Waltham, MA, United States). The steady-state and time-resolved photoluminescence (PL) spectra were tested with an Edinburgh FLSP980 fluorescence spectrometer (Edinburgh Instruments Ltd., Livingston, UK). The cyclic voltammetry (CV) curves were tested using a CH Instruments CHI760E electrochemical work station (CH Instruments Inc., Austin, TX, USA). The nitrogen adsorption−desorption isotherms were measured with a Quantachrome Auosorb-iQ2-MP analyzer (Quantachrome Instruments Inc., Boynton Beach, FL, USA). X-ray photoelectron spectroscopy (XPS) were performed in an ultrahigh vacuum chamber (ESCALAB 250Xi) using an XR6 monochromated AlKα X-ray source (hν = 1486.6 eV) with a 900 mm spot size (Thermo Fisher Scientific Inc., Waltham, MA, USA). The H_2_ yields were tested with a Fuli GC9790 II gas chromatography (Fuli Instruments Inc., Zhejiang, China). The apparent quantum yields (AQYs) were achieved using a monochromatic LED light source (Zolix, MLED4-1/M425, λ = 425 nm), and the photon flux was ascertained to be 1083 μmol h^−1^ (Zolix Instruments CO., LTD, Beijing, China). The Pd contents in all of the samples were obtained by inductively coupled plasma-atomic emission spectrometry (ICP-AES; SPECTRO CIROS VISION, spectra range: 120–800 nm, and holographic grating: 2924 line/mm) (SPECTRO Analytical Instruments, Kleve, Germany).

### 2.3. Synthesis and Preparation

#### 2.3.1. Synthesis of MOC-Q2

The solution of Pd(BF_4_)_2_(MeCN)_4_ (44 mg, 0.1 mmol, 1 eq) in DMF (1 mL) was added into a stirring solution of L-2 (144 mg, 0.2 mmol, 2 eq) in DMF (2 mL), and the resulting solution was stirred at room temperature for 10 min and filtered. Then, the product was precipitated by adding diethyl ether, and it was washed with ethyl acetate and dried in a vacuum to give 118 mg MOC-Q2 as a brown solid (yield: 63%), which was in accordance with our previous work [32].

#### 2.3.2. Preparation of g-C_3_N_4_

First, 10 g of urea was put in a 50 mL crucible and calcinated in a muffle furnace at 550 °C for 2 h at a heating rate of 10 °C/min. Then, the obtained solid was ground into powder and treated at 500 °C for another 2 h [30].

#### 2.3.3. Preparation of g-C_3_N_4_/MOC-Q2

The composite material g-C_3_N_4_/MOC-Q2 was prepared using a solution dipping method. First, 2 mg MOC-Q2 was dissolved in 100 μL DMSO. Then, 50 mg g-C_3_N_4_ was ultrasonically dispersed in 10 mL acetonitrile for 1 h. Thereafter, different volumes of MOC-Q2 solution were added dropwise into the above g-C_3_N_4_ suspension, i.e., 7.5, 17.5, 25.0, and 50.0 µL. For keeping the same total volume of the mixtures, 42.5, 32.5, and 25.0 µL of DMSO was subsequently added to the above mixtures, respectively. Afterward, the mixture was vigorously agitated for 12 h at room temperature, followed by evaporation and vacuum drying. Finally, resulting samples with MOC-Q2 mass fractions of 0.3, 0.7, 1.0, and 2.0 wt% were obtained.

#### 2.3.4. Preparation of g-C_3_N_4_/L-2 (0.7 wt%)

The preparation method was the same as that of g-C_3_N_4_/MOC-Q2 (0.7 wt%), except for replacing MOC-Q2 with L-2.

#### 2.3.5. Preparation of Pd/g-C_3_N_4_/L-2 (0.7 wt%)

Pd nanoparticles were loaded onto g-C_3_N_4_/L-2 (0.7 wt%) using a photo-deposition method. The calculated amounts of H_2_PdCl_4_ corresponding to 0.7 wt% were added into a g-C_3_N_4_/L-2 (0.7 wt%) suspension and then stirred for 2 h under 300 W Xenon lamp irradiation. Then, the mixture was centrifuged to obtain the grey powder, which was washed with deionized water and dried under a vacuum to gain product.

### 2.4. Photocatalytic H_2_ Generation

The photocatalytic H_2_ generation experiments were performed in a closed quartz reactor. In detail, 5 mg of catalyst powder was added into a quartz reactor containing 2 mL triethanolamine (TEOA) and 18 mL water. The mixture was ultrasonic-dispersed and bubbled with N_2_ for 30 min to eliminate O_2_. Then, the reactor was irradiated with a Xenon lamp (300 W, λ ≥ 420 nm), and the amounts of produced H_2_ were detected by gas chromatography (GC). Especially, for 5 mg MOC-Q2, considering its poor solubility in the aqueous phase, it was dissolved in a mixed solution containing 16 mL DMSO, 2 mL water, and 2 mL TEOA.

### 2.5. Apparent Quantum Yield (AQY) Measurements for H_2_ Evolution

The AQY measurements were performed in a closed quartz reactor. In detail, 5 mg of catalyst powder was added into a quartz reactor containing 4.5 mL water and 0.5 mL TEOA. The mixture was ultrasonic-dispersed and bubbled with N_2_ for 30 min to eliminate O_2_. Then, the reactor was irradiated with a 425 nm monochromatic LED, and the amounts of produced H_2_ were detected by GC. The AQYs were measured with the following formula:$$\Phi =\frac{\mathrm{number}\;\mathrm{of}\;\mathrm{transfered}\;\mathrm{electrons}}{\mathrm{number}\;\mathrm{of}\;\mathrm{incident}\;\mathrm{photons}}\times 100\%=\frac{2\times n(H_{2})}{n\left(\mathrm{photons}\right)}\times 100\%$$

### 2.6. Hydroxyl Radical Trapping Experiment

First, 20 mg of g-C_3_N_4_, MOC-Q2 or g-C_3_N_4_/MOC-Q2 (0.7 wt%) was dispersed in a sealed 40 mL quartz glass bottle containing 10 mL aqueous solution containing 5 × 10^−4^ M terephthalic acid (TA) and 2 × 10^−3^ M NaOH. Then, the reactor was irradiated with a Xenon lamp (300 W, λ ≥ 420 nm) for 1 h. After the reaction, the mixture was filtered through an aqueous phase membrane filter, and the filtrate was used for the fluorescence measurement.

## 3. Results and Discussion

### 3.1. Synthesis of MOC-Q2 and g-C_3_N_4_/MOC-Q2

The synthesis of MOC-Q2 and the preparation of hybrid materials g-C_3_N_4_/MOC-Q2 and Pd/g-C_3_N_4_/L-2 are given in the Experimental Section. The successful formation and the geometric configuration of MOC-Q2 was confirmed in our previous study [32]. The actual MOC-Q2 contents of the prepared samples g-C_3_N_4_/MOC-Q2 confirmed by ICP-AES are listed in Appendix A.

### 3.2. Characterization of g-C_3_N_4_/MOC-Q2

The PXRD patterns of g-C_3_N_4_, MOC-Q2, and the hybrid samples g-C_3_N_4_/MOC-Q2 are depicted in Figure 1. The pristine g-C_3_N_4_ and g-C_3_N_4_/MOC-Q2 materials displayed similar characteristic peak of 13.1° and 27.7°, demonstrating the (100) and (002) crystal plane of g-C_3_N_4_. Moreover, when MOC-Q2 contents gradually increased in the hybrids, the (002) peak of the g-C_3_N_4_/MOC-Q2 (0.3/0.7/1.0/2.0 wt%) samples slightly shifted to a lower angle. According to the Bragg formula, the decrease in the angle of the (002) peak indicated the interlayer spacing of g-C_3_N_4_/MOC-Q2 became larger, which can be attributed to the successful incorporation of MOC-Q2 into g-C_3_N_4_.

The SEM and TEM images displayed the stacked nanosheet structure of the g-C_3_N_4_/MOC-Q2 (Figure 2a–d). Owing to the tiny size of MOC-Q2, scarce signals related to MOC-Q2 were detected in the spectrogram. However, from the element mapping results of g-C_3_N_4_/MOC-Q2 (Figure 2e,f), it could be seen that the S and Pd elements, the exclusive elements of MOC-Q2, were detective all over the sample, thus confirming the homogeneously scattering of MOC-Q2 in the composite material.

The nitrogen adsorption−desorption tests of g-C_3_N_4_ and g-C_3_N_4_/MOC-Q2 (0.7 wt%) samples were carried out at 77 K. The isotherms are shown in Appendix A, and the BET surface areas and pore volumes are presented in Appendix A. As shown in Appendix A, g-C_3_N_4_ and g-C_3_N_4_/MOC-Q2 (0.7 wt%) exhibited H3 type hysteresis loops, which agreed with the layered structure of g-C_3_N_4_ materials. Furthermore, the pore size distribution diagram in Appendix A showed that there were many mesopores, and a few micropores and macropores in g-C_3_N_4_. As for the g-C_3_N_4_/MOC-Q2 (0.7 wt%), there were a large number of mesopores and a small amount of macropores, but the micropores belonging to g-C_3_N_4_ disappeared. The specific surface area and pore volume in g-C_3_N_4_/MOC-Q2 (0.7 wt%) was 54.5 cm^2^ g^−1^ and 0.33 cm^3^ g^−1^, respectively, both smaller than those of g-C_3_N_4_, indicating that some pores in g-C_3_N_4_ were blocked by MOC-Q2.

The UV−VIS diffuse reflection spectra of g-C_3_N_4_, MOC-Q2, and a series of g-C_3_N_4_/MOC-Q2 samples are illustrated in Figure 3. g-C_3_N_4_ showed a strong UV absorption peak with an edge of ca. 457 nm, related to its 2.71 eV optical band gap according to the Tauc plot (Appendix A). MOC-Q2 had a wide absorption region from 300 to 600 nm, owing to its effective π−π* transition. As for the g-C_3_N_4_/MOC-Q2 composites, superimposed absorption bands of g-C_3_N_4_ and MOC-Q2 were observed. As the MOC-Q2 contents increased, the absorption intensity at a range of 400–600 nm was gradually enhanced.

To explore the interaction between MOC-Q2 and g-C_3_N_4_ in the composite material, the FT-IR spectra were measured as shown in Figure 4. The absorption peaks of MOC-Q2 were found in the region of 700 to 1600 cm^−1^. The peaks at 1637, 1573, 1460, 1407, 1316, and 1240 cm^−1^ in g-C_3_N_4_ represented the stretching vibrations of the triazine rings, while the peak at 812 cm^−1^ was a characteristic absorption that came from the deformation vibration of triazine rings. The g-C_3_N_4_/MOC-Q2 (2 wt%) showed some typical peaks of g-C_3_N_4_ as well, but the deformation vibration peak of the triazine ring shifted to 814 cm^−1^, which might be due to the π−π stacking effect that existed between MOC-Q2 and g-C_3_N_4_. In addition, the broadened and weakened peaks in the range from 3000 to 3500 cm^−1^ in the g-C_3_N_4_/MOC-Q2, which were ascribed to the amine stretching vibrations of g-C_3_N_4_, demonstrated that hydrogen bonds formed between the uncoordinated pyridine moieties in MOC-Q2 and the amine groups on g-C_3_N_4_.

For testifying the electron transfer feasibility of this system, the energy levels of the constituents were measured. In the CV curve of MOC-Q2 in Appendix A, the onset oxidation potential (E_ox_) of MOC-Q2 appearing at 0.97 V vs. NHE, which corresponded to the Highest occupied molecular orbital (HOMO) energy level, was more positive than the CB edge potential (−1.03 V vs. NHE) of g-C_3_N_4_ [31], allowing for electron transfer from the CB edge of g-C_3_N_4_ to the oxidized MOC-Q2 so as to regenerate the neutral MOC-Q2. As seen from the normalized absorption and emission spectra of MOC-Q2 in Appendix A, its E_0-0_ transition energy was determined to be 2.58 eV from the intersection (Appendix A). Therefore, the excited state potential, which corresponded to its Lowest unoccupied molecular orbital (LUMO) level, was calculated to be −1.61 V vs. NHE judging by its E_ox_ and E_0-0_ value. The reduction potential of H^+^ was more positive than the LUMO value of MOC-Q2, so that MOC-Q2 could conduct a proton reduction reaction via the Pd^2+^ catalytic sites. On the other hand, the VB edge potential (1.68 V vs. NHE) of g-C_3_N_4_ [31] was more positive than that of the sacrificial agent TEOA (0.84 ± 0.12 V vs. NHE) [33] and therefore it is thermodynamically feasible for the efficient supply of electrons from TEOA to the positive-charged VB of g-C_3_N_4_ during the reaction.

### 3.3. Photocatalytic H_2_ Evolution

The photocatalytic H_2_ evolution experiments were performed under visible light (λ > 420 nm) irradiation in an aqueous solution with triethanolamine (TEOA) as the sacrificial agent. As shown in Figure 5 and Appendix A, when the content of MOC-Q2 on the g-C_3_N_4_ increased from 0.3 to 0.7 wt%, the photocatalytic activity enhanced from 2693 to 6423 µmol g^−1^ h^−1^ in 5 h. For g-C_3_N_4_/MOC-Q2 (0.7 wt%), the initial turnover number (TON) reached 20,955 based on the MOC-Q2 amounts, or 10,478 based on the number of Pd atoms. However, further increasing the incorporation amounts of MOC-Q2 conversely lowered the activity. For the g-C_3_N_4_/MOC-Q2 (1.0 and 2.0 wt%) samples, H_2_ yielded rates of 3153 and 2953 µmol g^−1^ h^−1^, respectively. We speculated that excessive amounts of MOC-Q2 would cause the accumulation of MOC-Q2 on the g-C_3_N_4_ surface, which resulted in intermolecular quenching, hindered the electron transfer efficiency, and thus decreased the catalyst activity. In addition, all the g-C_3_N_4_/MOC-Q2 samples exhibited a much higher H_2_ generation activity than the bare MOC-Q2 used in the homogeneous solution (855 µmol g^−1^ h^−1^), implying that the heterostructure of g-C_3_N_4_/MOC-Q2 could improve the effective charge transfer. As controls, Pd/g-C_3_N_4_/L-2 (0.7 wt%) samples with the same element composition were prepared and tested. The hydrogen production rate of Pd/g-C_3_N_4_/L-2 (0.7 wt%) was determined to be 156 µ mol g^−1^ h^−1^, far lower than that of g-C_3_N_4_/MOC-Q2 (0.7 wt%), indicating that MOC-Q2 played an essential role in the photocatalytic process, in which the ligand-to-metal charge transfer transition between L-2 and Pd^2+^ occurred.

To explore the durability of the MOC-Q2/g-C_3_N_4_ system, two reaction cycles were performed (Figure 6). The H_2_ yield in the second cycle decreased slightly, which might be due to the desorption or dissociation of a small part of MOC-Q2 during the photocatalytic reaction. A total H_2_ yield of 60.82 mmol g^−1^ was obtained, corresponding to a total TON_[MOC-Q2]_ of 39,695 and TON_[Pd]_ of 19,847, which was the highest of all of the reported MOC-based PMD under visible light irradiation (Table 1).

### 3.4. Mechanism Study

XPS measurements were performed to monitor the chemical states of Pd before and after the photocatalytic reaction (Appendix A). Considering that the Pd contents in g-C_3_N_4_/MOC-Q2 (0.7 wt%) were too low to be detected, g-C_3_N_4_/MOC-Q2 (2 wt%) with high Pd contents was used for the test. For the fresh sample, two peaks located at 337.7 and 342.8 eV with an area ratio of 3:2 could be found, which could be attributed to 3d_5/2_ and 3d_3/2_ orbits of Pd^2+^, respectively. After reaction for 10 h, the 3d_5/2_ and 3d_3/2_ peaks of Pd^2+^ shifted to 336.7 and 341.9 eV. In addition, two new peaks appeared at 334.8 and 340.1 eV with an area ratio of 3:2, representing the 3d_5/2_ and 3d_3/2_ orbits of Pd^0^. These changes indicated that the break of Pd-N took place during the photocatalytic process, resulting in the formation of simple substance Pd^0^ (Pd black), which was in accordance with our previous report [31]. In addition, the XRD patterns of the spent sample showed a negligible change compared with that of the fresh sample, testifying that the crystal form of the sample did not change during the reaction (Appendix A). In addition, the isolated feature of Pd on the spent g-C_3_N_4_/MOC-Q2 (2 wt%) sample after the photocatalytic reaction was observed by high-resolution TEM images (Appendix A).

To verify the electron transfer properties in g-C_3_N_4_/MOC-Q2, the steady-state and time-resolved PL spectra of g-C_3_N_4_, MOC-Q2, and g-C_3_N_4_/MOC-Q2 (0.7 wt%) were tested. As shown in Appendix A, the bare g-C_3_N_4_ exhibited strong fluorescence centered at 465 nm under 360 nm excitation, and MOC-Q2 showed trace fluorescence. Whereas for g-C_3_N_4_/MOC-Q2 (0.7 wt%), a fluorescence peak at 465 nm, but with a significantly decreased intensity compared to g-C_3_N_4_, was observed. This result suggested that some photogenerated electrons in g-C_3_N_4_ transferred to MOC-Q2, and thus led to the reduced fluorescence intensity of g-C_3_N_4_/MOC-Q2. The fitted fluorescence lifetime from the time-resolved PL spectra (Appendix A) were 10.1 ns for g-C_3_N_4_, 9.5 ns for g-C_3_N_4_/MOC-Q2 (0.7 wt%), and 1.2 ns for MOC-Q2, consistent with the steady-state PL results.

It is very important to confirm the Z-scheme mechanism for the heterojunction of g-C_3_N_4_/g-C_3_N_4_. In the presence of H^+^, the O_2_ in the reaction mixture can be reduced by the electron on the CB of g-C_3_N_4_ (Figure 7a), and the specific processes are listed as Equations (1)–(5) [49].
g-C_3_N_4_ + hv → h_VB_^+^ + e_CB_^−^(1)
e_CB_^−^ + O_2_ → O_2_•^−^(2)
O_2_•^−^ + H^+^ → HO_2_•(3)
e_CB_^−^ + HO_2_• + H^+^ → H_2_O_2_(4)
H_2_O_2_ + e_CB_^−^ → •OH + OH^−^(5)

Herein, we carried out the experiments of capturing •OH free radicals by terephthalic acid (TA), because TA could quickly combine with •OH to form a strongly fluorescent compound TA-OH, which has been widely used to testify the Z-scheme mechanism. As seen from Figure 8, the reaction mixture of the g-C_3_N_4_ sample with TA presented strong fluorescence at 430 nm, using 315 nm as the excitation wavelength, while that of the MOC-Q2 sample with TA had no fluorescence. As for the reaction mixture of the g-C_3_N_4_/MOC-Q2 (0.7 wt%) sample with TA, fluorescence with a medium intensity was observed. Considering that the produced •OH in the reaction mixture could only be generated by the electrons in the CB of g-C_3_N_4_, the attenuation of the fluorescence intensity came from the incorporation of MOC-Q2, which accepted some electrons in the CB of g-C_3_N_4_ and then reduced the yields of •OH (Figure 7b). Thus, it was verified that the as-prepared g-C_3_N_4_/MOC-Q2 system worked under a Z-scheme electron transfer route in the photocatalytic reaction. Therefore, we reasonably speculated the working mechanism of this system as follows: under visible light irradiation, g-C_3_N_4_ and MOC-Q2 synchronously entered an excited state and formed photogenerated electron−hole pairs. Then, the photo excited electrons on the CB of g-C_3_N_4_ combined with the holes on the HOMO of MOC-Q2, while the electrons on the LUMO of MOC-Q2 and the holes on the VB of g-C_3_N_4_ were retained to participate in the external reduction/oxidation reactions, respectively (Figure 9).

## 4. Conclusions

In summary, we designed and synthesized a photosensitive tridentate ligand L-2, and used four L-2 ligands with two catalytic Pd^2+^ centers to construct a M_2_L_4_-type molecular cage MOC-Q2. The as-prepared MOC-Q2 featured visible-light-harvesting and photo-induced LMCT transition properties, which allowed it to serve as a PMD for visible-light-driven water splitting. To overcome the instability of the homogeneous MOC catalyst and further improve the performances, we combined MOC-Q2 with g-C_3_N_4_ to prepare a direct Z-Scheme composite material g-C_3_N_4_/MOC-Q2. The optimized g-C_3_N_4_/MOC-Q2 (0.7 wt%) catalyst achieved a remarkable H_2_ generation activity of 6423 µmol g^−1^ h^−1^ and a TON value of 39,694 based on MOC amounts under visible light, far outperforming the homogenous MOC-Q2. The increased performance of the g-C_3_N_4_/MOC-Q2 catalyst could be owing to the superior charge-separation property of the Z-Scheme heterojunction, as well as the effective stabilizing effect through heterogenizing MOC-Q2 PMD with the polymeric g-C_3_N_4_ matrix. This research provides a promising idea to develop heterogeneous MOC-based catalysts for efficient and persistent photocatalytic water splitting.

## Data Availability

The data presented in this study are available in this article.

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
