# Peer review of "Direct Z-Scheme Heterojunction Catalysts Constructed by Graphitic-C3N4 and Photosensitive Metal-Organic Cages for Efficient Photocatalytic Hydrogen Evolution"

_nanomaterials, 2022, doi:10.3390/nano12050890_

Round 1

Reviewer 1 Report

The manuscript describes synthesis of the composite photocatalysts based on g-C3N4 and metal-organic cages (MOCs). Synthetic approaches look quite new and original, the work is well structured, photocatalysts are characterized by modern methods. I have only a few minor comments.

  1. Synthetic procedures should be described at least in a few words, links to previous publications of the authors are not enough.
  2. The article has problems with references and automatic numbering of figures.
  3. The absorption edge calculations (Kubelka-Munk transformations).
  4. Fig. 5. The error of measurements should be shown.
  5. The deactivation of the samples was observed. What is the reason of this phenomenon. XRD and XPS investigations after the photocatalytic tests should be carried out.
  6. Table S5 should be moved to the main text, because the comparison of the activities is very important.

Author Response

Dear Editor,

Thanks very much for your comments. We have revised our manuscript according to the reviewers’ comments. The detailed responses to the comments are listed in the uploaded files.

Thank you for your kind consideration.

Sincerely yours,

Jun-Min Liu

Reviewer 2 Report

This manuscript describes photocatalytic hydrogen (H2) evolution using heterogeneous catalysts consisting of graphitic C3N4 and metal-organic cage (MOC) hybrids. Overall, I think this manuscript address decent progress in this field so I would support the publication of this manuscript in Nanomaterials. Yet, I would be more than happy if the following issues are resolved before the Editor makes any decision.

(1) A lot of syntax(format) errors

This manuscript contains syntax errors, represented by “Error! Reference source not found”. This makes it difficult to focus on the manuscript. The authors should tidy up the manuscript. 

(2) Insufficient experimental details

The authors should provide the experimental details for future readers. For example, Section 2.3.1. and Section 2.3.2.

(3) The oxidation state of Pd in g-C3N4/MOC-Q2

First, this reviewer would suggest switching the location of Figure S3(A) and Figure S3(B). According to the XPS results using Pd 2wt% (instead of 0.7 wt%), it seems that the isolated Pd2+ would be agglomerated to form metallic Pd (Pd0+). Such an agglomeration needs to be tested by ex-situ STEM measurement before and after a 10 h photocatalytic test. To verify if the suggested mechanism is valid in Pd 0.7 wt%, the same measurement needs to be done together.

(4) High-resolution STEM measurement

Along with the question above, it would be better to provide high-resolution STEM images to capture the isolated feature of Pd in MOC.

(5) Scheme 1

Scheme 1 is one of the important figures in this manuscript. Despite its importance, this reviewer is not satisfied with its quality. Scheme 1 needs to be improved in terms of readability and visualization.

(5) The authors may want to quote the following references as they resolved the issues in this field.

  1. a) g-C3N4: E. W. Shin et al., Nanomaterials, 2022, DOI: 10.3390/nano12020179
  2. b) Ag@Au Nanoprisms/g-C3N4: X. Guo et al., Nanomaterials, 2021, DOI: 10.3390/nano11123347
  3. c) Pt-g-C3N4: J. Zhang et al., Nanomaterials, 2021, DOI: 10.3390/nano11123266
  4. d) Pd-based catalysts: J. G. Chen et al., Acc. Chem. Res., 2020, DOI: 10.1021/acs.accounts.0c00277
  5. e) Thiophene-doped g-C3N4: H. Li et al., Chinese J. Catal., 2021, DOI: 10.1016/S1872-2067(20)63674-9

Author Response

(The authors gave the same response as above.)
